# CleanUpRNAseq: An R/Bioconductor Package for Detecting and Correcting DNA Contamination in RNA-Seq Data

**DOI:** 10.3390/biotech13030030

**Published:** 2024-08-03

**Authors:** Haibo Liu, Kai Hu, Kevin O’Connor, Michelle A. Kelliher, Lihua Julie Zhu

**Affiliations:** 1Department of Molecular, Cell and Cancer Biology, University of Massachusetts Chan Medical School, 364 Plantation Street, Worcester, MA 01605, USA; haibo.liu@umassmed.edu (H.L.); kai.hu@umassmed.edu (K.H.); michelle.kelliher@umassmed.edu (M.A.K.); 2Department of Molecular Medicine, University of Massachusetts Chan Medical School, 364 Plantation Street, Worcester, MA 01605, USA; 3Department of Genomics and Computational Biology, University of Massachusetts Chan Medical School, 364 Plantation Street, Worcester, MA 01605, USA

**Keywords:** RNA-seq, DNA contamination, R package, bioinformatics, GC bias, gene expression

## Abstract

RNA sequencing (RNA-seq) has become a standard method for profiling gene expression, yet genomic DNA (gDNA) contamination carried over to the sequencing library poses a significant challenge to data integrity. Detecting and correcting this contamination is vital for accurate downstream analyses. Particularly, when RNA samples are scarce and invaluable, it becomes essential not only to identify but also to correct gDNA contamination to maximize the data’s utility. However, existing tools capable of correcting gDNA contamination are limited and lack thorough evaluation. To fill the gap, we developed CleanUpRNAseq, which offers a comprehensive set of functionalities for identifying and correcting gDNA-contaminated RNA-seq data. Our package offers three correction methods for unstranded RNA-seq data and a dedicated approach for stranded data. Through rigorous validation on published RNA-seq datasets with known levels of gDNA contamination and real-world RNA-seq data, we demonstrate CleanUpRNAseq’s efficacy in detecting and correcting detrimental levels of gDNA contamination across diverse library protocols. CleanUpRNAseq thus serves as a valuable tool for post-alignment quality assessment of RNA-seq data and should be integrated into routine workflows for RNA-seq data analysis. Its incorporation into OneStopRNAseq should significantly bolster the accuracy of gene expression quantification and differential expression analysis of RNA-seq data.

## 1. Introduction

Since its debut in 2006, RNA sequencing (RNA-seq) has surged in popularity as a go-to technology for transcriptome analysis and broader studies of RNA biology across diverse organisms, ranging from viruses and prokaryotes to eukaryotes [1,2,3,4,5]. The initial material for RNA-seq (hereafter referred to as conventional bulk RNA-seq unless stated otherwise) is typically total RNA, consisting of over ten distinct RNA species with widely varying abundance. These include protein-coding messenger RNA (mRNA), noncoding RNA (ncRNA) such as ribosomal RNA (rRNA), transfer RNA (tRNA), microRNA (miRNA), and small nucleolar RNA (snoRNA). Among these RNA species, rRNA is the most abundant, constituting 80% to 90% of total RNA by mass, followed by tRNA (10% to 15%) [6,7,8]. However, both rRNA and tRNA are often of little or no interest to most researchers. Fortunately, tRNAs are seldom sequenced by commonly used RNA-seq protocols because of their abundant base modifications and stable, compact secondary and tertiary structures [9,10]. Consequently, a crucial step in most RNA-seq protocols is to exclude rRNA from sequencing to enhance sequencing efficiency, improve sensitivity, and reduce costs. This goal can be achieved through the selection of polyA(+)-RNA or rRNA depletion strategies. The choice between the two strategies depends on the organisms of interest, research goals, and RNA integrity. For high-quality eukaryotic RNA samples, polyA(+)-RNA, mainly mRNA and long ncRNA, can be captured using oligo-dT-coated magnetic beads before reverse transcription [1] or selectively reverse-transcribed with oligo-dT primers [11,12,13]. On the other hand, various rRNA depletion methods are available for low-quality eukaryotic RNA samples, such as those from formalin-fixed, paraffin-embedded (FFPE) tissues [14], and RNA from prokaryotes [15]. These methods include (1) hybridization with biotinylated antisense oligonucleotide probes against rRNA, followed by streptavidin-coated magnetic bead pull-down [16,17]; (2) hybridization with antisense oligonucleotide probes against rRNA, followed by RNase H digestion and subsequent DNase I digestion [18]; and (3) blocking reverse transcription of rRNA with locked nucleic acid probes complementary to rRNA [19]. Although most RNA-seq experiments for eukaryotes have historically used polyA(+)-RNA selection methods, RNA-seq via rRNA depletion offers broader applications, i.e., allowing sequencing of non-polyadenylated RNA molecules, including a small fraction of mRNA, a majority of ncRNA in eukaryotes, and prokaryotic RNA.

Besides the highly abundant but often uninformative rRNA, genomic DNA (gDNA) contamination presents another challenge in RNA-seq. This occurs because of the presence of co-extracted gDNA during RNA preparation [20,21]. If not thoroughly removed, residual gDNA can be carried over into the sequencing library via the following mechanisms: (i) gDNA is replicated by reverse transcriptase during reverse transcription, albeit less efficiently than RNA. This is because reverse transcriptase acts as a both DNA-dependent and RNA-dependent DNA polymerase [22]; (ii) the replicated gDNA products, along with cDNA fragments, undergo indiscernible end-repair, phosphorylation, A-tailing, sequencing adaptor ligation, and PCR amplification. Consequently, the gDNA can be sequenced alongside RNA by RNA-seq [22], potentially leading to confounding signals. In fact, several studies have detected gDNA contamination in RNA-seq data [23,24,25,26,27,28,29], with levels depending on the sequencing sites and protocols used in the RNA-seq library preparation (See Supplementary Information of [27]). Li et al. showed that gDNA contamination levels could range from 0.7% to 22.7% in rRNA depletion-based RNA-seq libraries of human samples [23]. Even RNA-seq data generated by large consortia may contain gDNA contamination. For example, in a study led by the SEQC/MAQC-III Consortium, DNA contamination was spotted in RNA-seq data generated from the commercial human reference RNA samples and in one sample from the SEQC neuroblastoma project [27]. Signal and Kahlke found gDNA contamination in certain RNA-seq datasets within the ENA database [26]. Notably, gDNA contamination not only reduces RNA-seq efficiency and sensitivity but also creates artificial gene expression signals, resulting in the misinterpretation of transcriptome composition and mis-quantification of gene expression [23,25,27,29]. Li et al. showed that relatively high concentrations of gDNA contamination (>1%) altered gene quantification and increased false discovery rates in differential expression analysis, especially for genes with low abundance, in RNA-seq data generated by the rRNA-depletion method [23]. Verwilt et al. [25] further pointed out that extremely low input RNA-seq protocols, such as SILVER-seq [30], can suffer more severely from DNA contamination, contributing to the mis-quantification and false discoveries of mono-exonic transcripts [25]. Given the mandatory use of the rRNA-depletion method in RNA-seq library preparation from FFPE clinical samples and prokaryotic samples, it is of practical significance to quickly detect gDNA contamination in these RNA-seq data. It is equally important to investigate how prevalent gDNA contamination is in RNA-seq data deposited in public repositories and enable researchers to quickly identify high-quality public RNA-seq data for re-analyses. Furthermore, there are situations where discarding gDNA-contaminated RNA-seq data and regenerating uncontaminated counterparts is prohibitively expensive or simply unattainable due to constraints on sample availability. In such scenarios, it is desirable to perform in silico correction of gDNA contamination prior to downstream analyses.

The percentage of reads mapping to intergenic regions (IR%) is a valuable metric for gauging gDNA contamination in RNA-seq data, even though some reads, classified as intergenic, might indeed originate from exonic regions of genes that remain incompletely annotated or unknown due to gaps in genome annotation. Despite its importance, post-alignment assessment of gDNA contamination has been largely underperformed. Several tools can be repurposed for this by leveraging the report statistics on read distribution across various genomic regions, including genes, introns, exons, and intergenic regions. Examples include the CollectRnaSeqMetrics module of the Picard Tools (https://broadinstitute.github.io/picard/ (access on 16 July 2024)), Qualimap [31], and ALFA [32]. However, options for correcting such contaminations are more limited, with only two unpublished software tools available, gDNAx [33] and SeqMonk [34]. It is worth noting that both tools have limitations. While gDNAx can be easily integrated into automated RNA-seq analysis pipelines, its correction method is rudimentary, primarily filtering intronic and intergenic reads [35], but inadequately addressing gDNA reads mapped to exons. SeqMonk assumes that the median read density in intergenic regions across a genome represents the level of gDNA contamination. This estimate is then utilized to infer the gene-level read/fragment counts originating from gDNA contamination, which are subsequently subtracted from the observed counts to yield the gDNA contamination-corrected counts [34]. However, integrating SeqMonk into automated pipelines for large-scale RNA-seq data analysis poses challenges because it is a Java-based desktop tool with graphical interfaces. Li et al. introduced a correction method similar to SeqMonk, suggesting correcting gDNA contamination by subtracting the mean FPKM (Fragments Per Kilobase of transcript per Million mapped reads) across all intergenic regions from the observed gene expression in FPKM for each gene [23]. Both SeqMonk and Li’s method assume a uniform distribution of reads originating from gDNA contamination. Moreover, neither of these methods has undergone formal evaluation for their correction performance in the context of gene expression quantification or differential expression analysis. Therefore, despite the existence of tools for detecting gDNA contamination in RNA-seq data, there is still a need for a comprehensive and rigorously evaluated tool capable of both detection and correction.

To address this need, we developed an R/Bioconductor package, CleanUpRNAseq. It generates various diagnostic plots to facilitate the detection of gDNA contamination in RNA-seq data. With CleanUpRNAseq, users can visualize summary mapping statistics of RNA-seq data across various genomic features such as introns, exons, intergenic regions, rRNA-encoding regions, and organellar genomes. It also allows users to examine sample-level gene expression distributions, the percentages of genes surpassing specified thresholds, gene expression correlation between sample pairs, and similarity and variability of gene expression profiles across the entire experiment. More importantly, CleanUpRNAseq provides three methods for gDNA contamination correction for unstranded RNA-seq data and one dedicated method for stranded RNA-seq data. Benchmarking against RNA-seq data with known levels of gDNA contamination and real-world RNA-seq datasets with gDNA contamination demonstrates the efficacy of these methods in correcting gDNA contamination in RNA-seq data.

## 2. Materials and Methods

### 2.1. Overview of the CleanUpRNAseq Package

CleanUpRNAseq is an open-source R/Bioconductor package dedicated to detecting and correcting gDNA contamination in RNA-seq data. Built on the foundation of existing R/Bioconductor packages, it harnesses the ensembldb [36] and BSgenome packages for storing genome annotation and genome sequence information, GenomicRanges [37] and plyranges [38] for manipulating genomic range data, Rsubread [39] for summarizing aligned reads, tximport [40] for merging Salmon [41] quantification results, ggplot2 [42] for visualization, and voom [43] for linear model-based gDNA contamination correction.

This package provides a comprehensive suite of functions for gDNA contamination detection, with a series of diagnostic plots. More importantly, CleanUpRNAseq offers three correction methods for unstranded RNA-seq data: the “Global” method assumes a uniform distribution of gDNA contamination, which is a re-implementation of the SeqMonk method in R; the “GC%” method accounts for the GC-content-bias effect on read distribution across genomic regions; and the “IR%” method utilizes a linear model with IR% as a numeric covariate. It also contains features to optionally filter out multi-exonic genes lacking splicing junction support and to remove lowly expressed genes in uncontaminated samples of the same condition. For stranded RNA-seq data, CleanUpRNAseq provides a straightforward yet effective method, leveraging library strandedness information to correct gDNA contamination.

### 2.2. Description of the CleanUpRNAseq Package

A complete CleanUpRNAseq analysis requires four types of input data: (i) a genome annotation file in the GTF format from the Ensembl Genome Browser; (ii) a reference genome sequence file in the FASTA format, also from the Ensembl Genome Browser, or a BSgenome object for the reference of interest; (iii) RNA-seq read alignment files in the BAM format, generated using the same genome reference and genome annotation file as specified in (i) and (ii); (iv) quantification files generated by Salmon [41] pseudoalignment of RNA-seq reads using the Ensembl transcriptome with the same decoy genome FASTA file as specified in (ii). The flow chart for CleanUpRNAseq-based analysis of unstranded RNA-seq data is depicted in Appendix A.

Initially, an EnsDb SQLite database is established from the provided Ensembl GTF file using the make_ensdb function, which is a wrapper for the ensDbFromGtf function from the ensembldb package. Subsequently, a BSgenome package is created and installed through a series of sequential steps: the generate_multifasta function splits a multi-fasta file into individual fasta files for each chromosome/scaffold; the generate_seed_file function generates a seed file required for BSgenome package creation; and the forge_install_BSgenome function leverages the forgeBSgenomeDataPkg function from the BSgenome package to create a BSgenome package and the devtools package to check, build, and install the BSgenome package. In case where a UCSC-style BSgenome package is available from the Bioconductor repository, the UCSC2Ensembl function facilitates its conversion to a Ensembl-style BSgenome. It is crucial to ensure strict concordance between the version and source of GTF and FASTA files utilized in this process and those employed for RNA-seq read alignment and quantification.

Annotation files in the Simplified Annotation Format (SAF) (https://subread.sourceforge.net/SubreadUsersGuide.pdf (accessed on 13 July 2024)) are prepared for various genomic features, such as coordinate-collapsed introns, exons, intergenic regions, rRNA exons, mitochondrial and/or chloroplast genomes, using the get_feature_saf function. Subsequently, aligned reads in BAM files for each sample are summarized with these genomic features in both SAF and GTF formats. This process is facilitated by the summarize_reads function, which is a wrapper for the featureCounts function from the Rsubread package [39,44]. The output generated by the summarize_reads function serves as input for detecting gDNA contamination through the execution of several functions: (i) the check_read_assignment_stat function, for generating stacked bar plots showing read assignment statistics; (ii) the check_read_distribution, for producing dot plots displaying percentages of reads distribution across various genomic features such as exons, introns, intergenic regions, rRNA-coding regions, and mitochondrial and/or chloroplast genomes; (iii) the check_sample_correlation function, for creating smooth scatter plots illustrating the gene expression correlation between each pair of samples; (iv) the check_expression_distribution function, for generating boxplots, density plots, and empirical cumulative distribution functions presenting sample-level gene expression distributions; (v) the check_expressed_gene_percentage function, for producing dot plots showing the percentage of expressed genes in each sample based on user-defined cutoffs in counts per million (CPM) or transcripts per million (TPM); and (vi) the exploratory_analysis function, for creating a hierarchically clustered heatmap plot showing pairwise expression profile similarities and PCA score plots illustrating sample variabilities. Transcript expression quantified by Salmon, a fast and bias-aware mapper, is summarized into gene-level expression using the salmon_res function, which is a wrapper for the tximport function from the tximport package.

For unstranded RNA-seq data, the intergenic region-by-sample matrix, output by the summarize_reads function with the intergenic regions annotated in SAF format, is utilized for gDNA contamination correction through three approaches. In essence, for each sample, coverage in fragments per base (FPB) is calculated for intergenic regions by dividing the fragment counts by the lengths, measured in base pair (bp), of the corresponding intergenic regions. The median FPB value for intergenic regions with non-zero counts serves as an estimate of the global gDNA contamination level for the “Global” correction method. Per-gene contamination is calculated by multiplying the sample-specific global gDNA contamination level and the gene length output by the salmon_res function. For the “GC%” method, the GC content of individual intergenic regions and genes is calculated using the calculate_region_gc and calculate_gene_gc functions, respectively. For each sample, a loess regression model is fitted with the FPB of intergenic regions with non-zero counts as the response variable and the GC content as the independent variable. The GC content in the range of 0% to 100% is divided into 20 equal-width bins, and intergenic regions are assigned to their corresponding GC bins based on their GC content. Coverages in FPB of intergenic regions within each GC bin are predicted by the fitted loess regression model. The median coverage in FPB of intergenic regions within each GC bin serves as GC bin-specific estimate of gDNA contamination. For a given sample, the GC content of individual genes is similarly binned as for the intergenic region. Per-gene contamination is calculated by multiplying the GC bin-specific estimate of gDNA contamination for that sample and the length of the gene output by the salmon_res function. For both methods, per-gene gDNA contamination is subtracted from the raw per-gene counts, resulting in a gDNA contamination-corrected count matrix. These methods are implemented as the global_correction and gc_bias_correction functions. The gDNA contamination-corrected count matrix can be directly used for differential expression analysis. Optionally, this corrected matrix can be further filtered to remove multi-exonic genes lacking splicing junction supports and genes lowly expressed in uncontaminated samples of the same condition. These additional filtering steps must be executed with caution, as some libraries might have a limited representation of splicing junction reads, potentially leading to increased false negatives in differential expression analysis. The third correction method involves using the percentage of reads mapped to the intergenic regions (IR%) as a covariate in linear models for gDNA contamination correction or generalized linear models for differential expression analysis. CleanUpRNAseq provides the IR_percent_correction function to obtain the IR%-adjusted expression matrix in log_2_CPM scale, leveraging the voom method from the limma package [43,45].

For stranded RNA-seq data, where the proportion of stranded reads exceeds 0.9 [26], gene expression is quantified twice: once with the expected library strandedness and once with the opposite strandedness. This can be achieved using the conventional alignment method, followed by gene-level read summarization or the light-weight pseudo-alignment method implemented in Salmon [41]. The gDNA-corrected count matrix is derived by subtracting the count matrix resulting from the quantification based on the opposite strandedness from the one obtained from quantification based on the correct library strandedness.

To enhance usability, we developed two wrapper functions: create_diagnostic_plots and correct_for_contamination. These functions streamline the generation of diagnostic plots and the application of selected correction methods.

### 2.3. RNA-Seq Datasets

An RNA-seq dataset (Accession number HRA001834, Dataset I) with predefined levels of gDNA contamination was downloaded from the repository of the Genome Sequence Archive for Human of National Genomics Data Center (https://ngdc.cncb.ac.cn/gsa-human/ (accessed on 13 July 2024)). This dataset, generated by Li et al., was specifically designed to study the effect of gDNA contamination on differential expression analysis, as detailed in Appendix A [23]. Briefly, total RNA and gDNA were separately extracted from a culture of a human HapMap lymphoblast cell line purchased from the Coriell Institute. One portion of total RNA underwent DNase I, while the other did not. Six aliquots (Samples A1–A6, Group “DNase I (−)”) of total RNA not treated with DNase I were prepared, with each aliquot containing 250 ng of nucleic acids. Meanwhile, gDNA and DNase I-treated RNA aliquots were mixed at varying ratios, so that gDNA accounted for 0% (Samples B1–B6, Group “0%”), 0.01% (Samples C1–C6, Group “0.01%”), 0.1% (Samples D1–D6, Group “0.1%”), 1% (Samples E1–E6, Group “1%”), or 10% (Samples F1–F6, Group “10%”) of the total 250 ng of nucleic acids in the resulting mixtures by mass. The samples in Group “0%” were considered as the control for all comparisons in differential gene expression analysis. For Samples A1–A3 through F1-F3, polyA(+)-RNA was enriched, followed by library preparation using the TruSeq RNA Library Prep Kit (Illumina, San Diego, CA, USA), whereas for Samples A4-A6 through F4-F6, rRNA was depleted using the RiboMinus^TM^ Eukaryote Kit for RNA-seq (Invitrogen, Waltham, MA, USA), followed by library construction using the TruSeq Stranded Total RNA Library Prep Kit (Illumina). Paired-end reads (2× 50-bases) were generated on an Illumina HiSeq2000 platform. Surprisingly, the Salmon mapping results revealed that the libraries for rRNA-depleted samples were unstranded, although the authors declared the use of a stranded RNA-seq library preparation kit. Detailed metadata of this RNA-seq dataset is available in Appendix A.

Another RNA-seq dataset (Accession number: GSE260697, Dataset II), with two heavily contaminated samples inadvertently generated during a study investigating the relapse in T-cell acute lymphoblastic leukemia (T-ALL), was downloaded from Gene Expression Omnibus (GEO). Briefly, human CD7^+^CD1A^+^ cells (actively cycling, chemosensitive, T-lineage-committed leukemic T cell precursors) and CD7^+^CD1A^−^ cells (cycle-restricted, chemoresistant, multi-lineage leukemia-initiating T-cell precursors) were sorted from T-ALL patient derived xenografts (PDX) from six mice [46]. RNA-seq assays were conducted using the SMART-Seq^®^ v4 Ultra^®^ Low Input RNA Kit for Sequencing (Takara Bio, Shiga, Japan). Paired-end reads (2× 151 bases) were generated on an Illumina NovaSeq 6000 platform, as depicted in Appendix A. Additional details on the second RNA-seq dataset are available in Appendix A and are partially described in the original publication [46].

### 2.4. Preprocessing, Alignment, and Quantification

The quality of raw reads was evaluated using FastQC (v0.11.9) (https://www.bioinformatics.babraham.ac.uk/projects/fastqc/ (accessed on 13 July 2024)). For Dataset I, no trimming was performed, as the read quality was high and adaptor sequences were barely detected. However, for Dataset II, Trimmomatic (v0.32) [47] was employed to trim adaptor sequences and remove low-quality bases at the 3′ ends due to a notable presence of adaptor sequences in a considerable portion of reads. The trimming option settings were as follows: ILLUMINACLIP:adapter.fa:2:25:7:1:true TRAILING:3 SLIDINGWINDOW:4:15 MINLEN:25. After adaptor trimming, the reads with length ≥ 25 bp were kept. Paired-end reads were aligned to the human reference genome GRCh38 (Ensembl Release 110) using STAR (v2.7.10a) [48] in a two-pass mode, with the matching GTF (Ensembl Release 110) as gene annotation. The specific parameter configurations for STAR alignment were as follows: --outFilterMultimapNmax 10 --outFilterScoreMinOverLread 0.15 --outFilterMatchNminOverLread 0.15 --alignSJoverhangMin 8 --twopassMode Basic --outSAMattributes all --outSAMstrandField intronMotif --outFilterIntronMotifs RemoveNoncanonical --alignSJDBoverhangMin 1 --outFilterMismatchNmax 999 --outFilterMismatchNoverReadLmax 0.1 --seedSearchStartLmax 50 --seedSearchStartLmaxOverLread 0.15 --alignIntronMin 20 --alignIntronMax 1000000 --alignMatesGapMax 1000000 --outFilterType BySJout --limitSjdbInsertNsj 2000000 --outSAMtype BAM Unsorted. The BAM files generated from STAR alignment were coordinate-sorted using SAMtools (v1.16) [35]. Subsequently, reads aligned to different genomic features were tallied using the summarize_reads function for each type of genomic features in SAF format.

### 2.5. Pseudo-Alignment and Quantification Using Salmon

Paired-end reads, with adaptor sequences trimmed as described earlier, were mapped to the human transcriptome, encompassing all coding and none-coding RNAs (Ensembl Release 110), decoyed with the human reference genome GRCh38 (Ensembl Release 110), using Salmon (v1.9.0) [41]. The specific parameter configurations for Salmon quantification were as follows: --seqBias --gcBias --posBias --softclip --softclipOverhangs --biasSpeedSamp 5 -l IU --validateMappings. Quantification results for individual samples were imported into R using the tximport function of the tximport package [40], resulting in a gene-by-sample count matrix, a gene-by-sample abundance (in TPM, transcript per million) matrix, and a gene-by-sample length matrix.

### 2.6. gDNA Detection and Correction, and Differential Expression Analysis

The gDNA contamination in RNA-seq data from each sample was assessed using CleanUpRNAseq. Subsequently, the gDNA contamination in the gene-by-sample count matrices from featureCounts or tximport was corrected using CleanUpRNAseq, following the procedure outlined earlier. The gene expression matrices, after correction for gDNA contamination, underwent exploratory analysis and differential expression analysis to confirm the effectiveness of the correction. To compare different correction methods, we conducted differential expression analyses using DESeq2 (v1.32.0) [49], with raw count matrices and gDNA contamination-corrected count matrices, respectively. Initially, genes with extremely low expression were filtered out from the gene-by-sample count matrices. Differentially expressed genes (DEGs) were determined based on a series of log_2_(Fold-Change) thresholds (0, 0.585, and 1) and the BH method-adjusted *p*-value ≤ 0.05. DEGs were visualized by volcano plots and Sankey plots. 

## 3. Results

The percentage of reads mapping to intergenic regions (IR%) serves as an important indicator of gDNA contamination in RNA-seq data [23]. Elevated IR% values in RNA-seq data, relative to uncontaminated samples processed under similar conditions, typically signify gDNA contamination. This contamination not only increases the IR%, but also distorts gene expression quantification [23,25]. As gDNA contamination intensifies, its impact on gene expression quantification increases through several discernible effects: (1) shifts in sample-level gene expression distributions towards higher values; (2) increased detection of genes expressed above a specified threshold; (3) diminished correlation in gene expression profiles between contaminated and uncontaminated samples; (4) increased dissimilarity and variability among gene expression profiles of samples across an experiment. Thus, detecting potential gDNA contamination in RNA-seq data necessitates a multifaceted approach. To address this, we implemented a comprehensive set of functions in CleanUpRNAseq, enabling visualization of the following parameters: (1) read mapping percentages across various genomic features, including genes, exons, introns, intergenic regions, rRNA exons, and organellar genome(s); (2) distributions of sample-level gene expression; (3) percentages of genes with expression levels above user-defined thresholds; (4) pairwise correlation of gene expression profiles; and (5) similarity and variability in gene expression profiles across an entire experiment.

To correct gDNA contamination in RNA-seq data, we devised three methods for unstranded RNA-seq data: the “Global” method, the “GC%” method, and the “IR%” method, along with one tailored method for stranded RNA-seq data (refer to the Materials and Methods section). We evaluated the functionalities using two datasets, as detailed in the RNA-seq datasets section, and presented the results below.

### 3.1. CleanUpRNAseq Possesses the Capability to Detect and Correct Significant gDNA Contamination in RNA-Seq Data Produced through the rRNA-Depletion Method

To demonstrate the functionalities of CleanUpRNAseq, we analyzed a RNA-seq dataset with predefined levels of gDNA contamination in the RNA samples (Dataset I) [23]. We chose this dataset because all RNA samples originated from the same cell culture extract, and the differences between groups of samples were well-defined. Variations among sample groups were solely attributed to whether RNA samples underwent DNase I treatment and the varying levels of deliberately added gDNA to the DNase I-treated RNA samples. We hypothesized that applying an effective correction method would result in the corrected gene expression profiles of samples with added gDNA more closely resembling those without, compared to the uncorrected gene expression profiles affected by gDNA contamination. This dataset serves as an ideal benchmark for evaluating the performance of correction methods in minimizing false positive rates.

When analyzing the RNA-seq data generated by the polyA(+)-RNA selection method, we observed that moderate gDNA contamination (up to 10%, 25 ng) in RNA samples has little effect on gene expression quantification, differential gene expression analysis, and read distribution across various genomic regions (Appendix A). Differential expression analysis detected a very small number of differentially expressed genes between samples of any given level of added gDNA and the control group (Appendix A). These observations indicate that adding gDNA to the DNase I-treated RNA samples had a minimal impact on the RNA-seq data generated with the polyA(+)-RNA selection method.

In contrast, we observed significant impacts across various metrics for the RNA-seq data generated by the rRNA-depletion method (Appendix A). Notably, even a minimal addition of gDNA (as low as 1%, 2.5 ng) led to a discernible distinction. These differences manifested in several aspects, such as elevated IR%, alterations in gene expression profiles and distributions, increases in the proportion of genes exhibiting expression levels above one CPM or TPM, as well as a declined correlation between samples without added gDNA and those with 1% or more added gDNA (Appendix A). The impact of introducing 10% of gDNA on gene expression profiles became evident through boxplots (Appendix A). These effects on gene expression profiles, with varying percentages of added gDNA, were also apparent in dendrograms showing hierarchical clustering of gene expression profiles and PCA plots illustrating the variability in gene expression profiles across all samples (Appendix A). Further validation of the influence of added gDNA on gene expression profiles was attained through differential expression analysis. The number of differentially expressed genes between samples with added gDNA and the control group increased notably when 1% or more of gDNA was added to the RNA samples (Figure 1 and Appendix A). These findings indicate that gDNA contamination exerts a significantly greater impact on gene expression profiles in the RNA-seq data generated with the rRNA-depletion method compared to those from the polyA(+)-RNA selection method, consistent with findings by Li et al. [23].

We next performed gDNA contamination correction using the three different methods for unstranded RNA-seq data. For the RNA-seq data generated with the polyA(+)-RNA selection method, density plots showing gene expression distributions revealed that none of the three methods significantly modified the gene expression distributions of samples at any levels of gDNA contamination (Appendix A). This is likely because gDNA contamination has minimal impact on the RNA-seq data generated using this method. In contrast, for the RNA-seq data generated using the rRNA-depletion method, all three methods demonstrated effectiveness. Among them, the “IR%” method induced the most significant correction in gene expression profiles, while the other two methods exhibited comparable performance (Appendix A). The correction effects were further validated through differential expression analyses conducted between the control group and samples with any given level of added gDNA, before and after correcting gDNA contamination.

For RNA-seq data generated using the rRNA-depletion method, applying the “IR%” correction method led to a significant decrease in the numbers of DEGs between the control group and the samples with 1% or 10% of gDNA added, while the “GC%” and “Global” correction methods showed moderate but comparable effects (Figure 1 and Appendix A). In contrast, for polyA(+)-RNA selection-based RNA-seq data, the numbers of DEGs were initially small even before correcting gDNA contamination (Appendix A). The application of any of the three correction methods resulted in no reduction in the numbers of DEGs (Appendix A). Taken together, the “IR%” method demonstrated the highest effectiveness, followed by the “GC%” and “Global” methods, in correcting gDNA contamination at 1% or higher in the rRNA depletion-based RNA-seq data, while none of the three correction methods worked in the polyA(+)-RNA selection-based RNA-seq data.

Another interesting observation is that RNA samples not treated with DNase I contains less than 2% of gDNA [23] (Appendix A), yet we detected a higher number of DEGs between them and the control group compared to the number observed between the group of RNA samples with 10% of gDNA and the control group in the RNA-seq data generated using the polyA(+)-RNA selection method (Appendix A). In addition, the gene expression profiles of this group of samples are significantly different from other samples (Appendix A). Since the addition of up to 10% of gDNA to RNA samples minimally affected the gene expression profiles determined by the polyA(+)-RNA selection method-based RNA-seq assays, we infer that the RNA samples not treated with DNase I likely have different RNA compositions from those treated with DNase I. The alteration in RNA compositions might have originated from the DNase I digestion process and/or the following washing step. The DEGs resulting from differences in RNA compositions cannot be reduced by the gDNA contamination correction methods (Figure 1 and Appendix A). 

### 3.2. CleanUpRNAseq Possesses the Capability to Detect and Correct Detrimental Levels of gDNA Contamination in RNA-Seq Data Generated by the Selective Reverse Transcription of polyA(+)-RNA Method with Ultra Low Input RNA

In addition to analyzing RNA-seq data generated from RNA samples deliberately contaminated with controlled levels of gDNA using both the polyA(+)-RNA selection and rRNA-depletion methods, we further validated CleanUpRNAseq’s capabilities by analyzing an additional RNA-seq dataset generated using the SMART-Seq^®^ v4 Ultra^®^ Low Input RNA Kit for Sequencing (Takara Bio, Shiga, Japan), which selectively reverse-transcribes polyA(+)-RNA from the total RNA [46]. We identified two samples, CD1A(−)_m2_1 and CD1A(−)_m3_1, exhibiting extremely high IR%, alongside four other samples—CD1A(+)_m2_1, CD1A(+)_m3_1, CD1A(+)_m3_2, and CD1A(−)_m4_1—with slightly elevated IR%, compared to the remaining eight samples (Figure 2A). The boxplots, density plots, and empirical cumulative distribution plots collectively revealed notable discrepancies in the gene expression profiles of the two samples, CD1A(−)_m2_1 and CD1A(−)_m3_1, compared to the remaining 12 samples. As the expression profiles of the remaining samples exhibited relatively smaller variability (Figure 2B,D), the distinct patterns observed in those samples strongly indicate heavy gDNA contamination in CD1A(−)_m2_1 and CD1A(−)_m3_1, with slight contamination in CD1A(+)_m2_1, CD1A(+)_m3_1, CD1A(+)_m3_2, and CD1A(−)_m4_1. The assertion was further supported by the markedly higher percentages of genes with expression levels above one CPM or one TPM in the heavily contaminated samples, along with mildly elevated percentages in the slightly contaminated samples (Figure 2E). Correlation analysis and smooth scatter plots provided additional evidence of gDNA contamination in these samples (Figure 2F). Hierarchical clustering and PCA revealed significant batch effects, highlighting stark difference in gene expression profiles between the two heavily contaminated samples and the rest of samples from the same batch (Figure 2G,H). To avoid the side effect of gDNA contamination, samples heavily contaminated with gDNA (CD1A(−)_m2_1 and CD1A(−)_m3_1) and the paired samples (CD1A(+)_m2_1 and CD1A(+)_m3_1) were excluded from the differential gene expression analysis in the original publication [46].

We proceeded to evaluate the efficacy of the three methods for addressing gDNA contaminations in this RNA-seq dataset. Our premise rested on a reasonable assumption that gene expression profiles of biological replicates should exhibit high similarity, and that the effectively corrected gene expression profile of an RNA sample contaminated with gDNA should closely resemble that of an uncontaminated RNA sample extracted from the same biological material. Notably, in this dataset, both RNA samples CD1A(−)_m3_2 and CD1A(−)_m3_1 originated from the same population of CD7^+^CD1A^−^ cells, with the former uncontaminated and the latter heavily contaminated by gDNA. Likewise, both RNA samples CD1A(+)_m3_2 and CD1A(+)_m3_1 were derived from the same CD7^+^CD1A^+^ cell population, with the former uncontaminated and the latter heavily contaminated by gDNA. Furthermore, the CD1A(−)_m2 and CD1A(+)_m2 cells were sorted from the same mouse PDX, as were the CD1A(−)_m3 and CD1A(+)_m3 cells. Hence, we expected that following an appropriate correction of gDNA contamination, the gene expression profiles of CD1A(−)_m3_2 and CD1A(−)_m3_1 would exhibit higher similarity. After examining the density plots of gene expression profiles before and after correction with the three methods (Figure 3), we concluded that the “IR%” method performed best, followed by the “GC%” and “Global” methods, which exhibited similar performance. This claim is further supported by the results of differential expression analyses between the CD1A(+) and CD1A(−) groups, using gene expression matrices before and after the correction of gDNA contamination with the three methods. The largest number of DEGs, particularly the down-regulated ones, were identified when none of the correction methods was applied (Figure 4A). This aligns with the expectations, given that the two heavily contaminated replicates belong to the CD1A(−) group, and the comparison involves CD1A(+) versus CD1A(−). Applying the “IR%” method resulted in the fewest DEGs, a number much closer to that observed in the corresponding analysis with contaminated samples excluded, whereas similar numbers of DEGs were observed following the application of “GC%” and “Global” correction methods (Figure 4A). Sankey plots and volcano plots displaying DEGs indicated a significant reduction in the number of down-regulated DEGs in the CD7^+^CD1A^+^ cells compared to the CD7^+^CD1A^−^ cells, following application of the “IR” correction method (Figure 4B and Figure 5), demonstrating the effectiveness of the “IR%” method in mitigating gDNA contamination within the CD1A(−) group. This conclusion was further confirmed by the hierarchical clustering analysis of gene expression profiles, both before and after the application of the three correction methods (Appendix A). Notably, only after employing the “IR%” method were the samples CD1A(−)_m3_2 and CD1A(−)_m3_1 closest to each other. Similarly, the samples CD1A(+)_m3_2 and CD1A(+)_m3_1 exhibited the closest similarity (Appendix A). Importantly, the biological replicates of CD1A(+) samples formed a distinct cluster, clearly separated from the cluster formed by the biological replicates of the CD1A(−) samples, but this distinction became evident only after applying the “IR%” correction method (Appendix A). The results of PCA of the gene expression profiles before and after applying the three correction methods were consistent with those of the hierarchical clustering analysis (Appendix A).

Taken together, CleanUpRNAseq demonstrated its ability to detect and correct gDNA contamination in the RNA-seq data generated using the SMART-Seq^®^ v4 Ultra^®^ Low Input RNA Kit for Sequencing (Takara Bio, Shiga, Japan), with the “IR%” correction method proving to be the most effective in mitigating such contamination.

## 4. Discussion

gDNA contamination is an inherent problem in RNA preparation due to the close resemblance between DNA and RNA in their physicochemical properties [20,21]. Failure to effectively remove gDNA contamination from RNA extraction might compromise the reliability of RNA-based gene expression quantification and differential gene expression analysis. The impact of gDNA contamination on gene expression quantification via RT-qPCR has been extensively studied, leading to the development of methods to prevent gDNA amplification or to correct gDNA contamination [50,51]. However, the implication of gDNA contamination in RNA preparation on gene expression quantification via RNA-seq remains relatively understudied [23,25], with some discussion primarily found in online forums [52,53,54,55,56]. For instance, a search with keywords “DNA contamination” and “RNA-seq” returned eight relevant posts in BioStar (https://www.biostars.org/ (accessed on 13 July 2024)) [57]. While a couple of tools have emerged to address gDNA contamination in RNA-seq data, their performance lacks rigorous validation through benchmarking studies and peer-reviewed evaluation [33,34]. This underscores the necessity for further research to comprehensively understand and mitigate the influence gDNA contamination on RNA-seq based gene expression quantification and differential gene expression analysis.

In response to the pressing need for rapid identification and remediation of gDNA contamination in both publicly available and in-house RNA-seq datasets, we developed the CleanUpRNAseq package. Our tool underwent rigorous testing using RNA-seq data generated by three distinct methods: polyA(+)-RNA selection from total RNA by oligo-dT-coated magnetic beads, rRNA-depletion from total RNA by RiboMinus Eukaryote Kit for RNA-Seq (Invitrogen), and selective reverse transcription of polyA(+)-RNA in the context of total RNA by the SMART-Seq^®^ v4 Ultra^®^ Low Input RNA Kit for Sequencing (Takara Bio, Shiga, Japan).

CleanUpRNAseq demonstrates remarkable efficacy in detecting problematic levels of gDNA contamination in the RNA-seq data generated with the rRNA depletion (1% threshold) method. It also exhibits good sensitivity in detecting lightly or heavily contaminated samples in RNA-seq data generated by the SMART-Seq^®^ v4 Ultra^®^ Low Input RNA Kit for Sequencing (Takara Bio, Shiga, Japan). However, CleanUpRNAseq did not detect obvious gDNA contamination in the RNA-seq data generated with the polyA(+)-selection method, even when up to 10% of gDNA was added to the DNase I-treated RNA for library preparation. Our findings, aligned with Li et al.’s [23], highlight a significantly different impact of gDNA contamination on RNA-seq data generated by the polyA(+)-RNA selection and rRNA depletion methods. The observation of a more pronounced effect of gDNA contamination on RNA-seq data generated by the rRNA-depletion method warrants the necessity of robust contamination detection and mitigation strategies for datasets derived from this method and similar ones.

In addition to the two primary methods for conventional bulk RNA-seq, numerous innovative RNA-seq techniques have emerged over the years. These include single-cell RNA-seq methods, spatially resolved RNA-seq methods, nascent RNA-sequencing methods, translating RNA-sequencing methods, and RNA-seq methods for investigating RNA interactome [5,58]. While all methods share common mechanisms in carrying over residual gDNA contamination into the sequencing library, a notable difference lies in the initial amount of residual gDNA present before reverse transcription and PCR amplification. For example, during polyA(+)-RNA selection, the vast majority of gDNA contamination is effectively removed through the selective capture by oligo-dT-coated magnetic beads and subsequent washing steps, distinguishing it from other methods lacking stringent RNA selection or gDNA exclusion process. As a consequence, only a trace amount of random DNA fragments are non-specifically bound by beads [59], with a small fraction of gDNA fragments containing long stretches of A’s being selectively captured by beads during DNA breathing, accelerated by the procedure of RNA denaturation at 60 °C [60]. Hence, the polyA(+)-RNA selection method notably produces a sequencing library with significantly reduced and non-uniform gDNA contamination. Aligned with this, only a small number of genes were consistently detected as differentially expressed between the control group and the sample groups with various levels of added gDNA in polyA(+)-RNA selection-based RNA-seq. Therefore, we anticipate a minimal impact of gDNA contamination on the integrity of data generated by RNA-seq methods, which typically involve selective removal of gDNA contamination. Conversely, the impact of gDNA contamination is considerably more profound and widespread in the RNA-seq data generated via both the rRNA-depletion and the selective reverse transcription of polyA(+)-RNA methods, as neither of them effectively removes gDNA contamination prior to reverse transcription.

In the CleanUpRNAseq package, we implemented three methods to correct gDNA contamination in unstranded RNA-seq data. The “Global” method estimates the global contamination based on median read coverage in non-zero counted intergenic regions, assuming a uniform distribution of gDNA-derived reads across the genome. However, this assumption may not hold true at very low contamination levels or when read distributions are skewed by biased PCR amplification. In light of the GC-content bias effect on PCR amplification and gene quantification in RNA-seq [61,62,63,64], we devised the “GC%” method, which estimates gDNA contamination for genes in each GC-content bin based on the median read coverage in intergenic regions within the same GC-content bin. Contrary to expectations, but in line with Li et al.’s [23], incorporating GC-content bias did not significantly improve contamination correction compared to the simpler “Global” method. The third method, “IR%” correction, incorporates the IR% as a covariate in linear models for correcting gDNA contamination or generalized linear models for differential expression analysis. It assumes a linear relationship between the IR rate and the log-transformed CPM of gene expression. This method harnesses the power of leveraging data from all samples to incorporate a global estimation of contamination as a covariate and model gene-specific effects. By employing this approach, it effectively addresses the limitations of pure global methodologies, resulting in an enhanced ability to estimate contamination effects and facilitate contamination removal, consistently outperforming others in mitigating problematic gDNA contamination levels for differential expression analysis.

However, when reads originating from gDNA contamination significantly deviate from a uniform distribution, such as in RNA-seq data with very low levels of gDNA contamination, or in instances of ‘jackpot’ effects caused by PCR amplification bias, or when a linear relationship between the IR rate and log-transformed CPM of gene expression is absent, none of the methods effectively address gDNA contamination in unstranded RNA-seq data. PolyA(+)-RNA selection-based RNA-seq results in non-uniform distribution due to the inefficient and selective capture of gDNA fragments containing stretches of As by oligo-dT-coated beads. In contrast, low contamination levels (<1%) in rRNA depletion-based RNA-seq disrupt uniform distribution due to sampling noise. However, correction is unnecessary when the contamination level is inconsequential, as contamination at such levels only minimally affects gene expression quantification and differential gene expression analysis. Therefore, correction for gDNA contamination is largely unnecessary for RNA-seq data generated with the polyA(+)-RNA selection method or those with gDNA contamination below 1% by mass.

For stranded RNA-seq data, we implemented a specialized method to address gDNA contamination, taking advantage of the stranded nature of the reads originating from RNA transcripts. Unlike single-stranded RNA transcripts, both strands of double stranded gDNA can serve as template for reverse transcriptase. Consequently, the percentages of sequenced gDNA reads derived from either strand should approach 50%. Our method for correcting gDNA contamination in stranded RNA-seq data operates on this premise, yet it assumes neither a uniform distribution of gDNA-derived reads nor a linear relationship between log-transformed CPM/count and gDNA contamination levels approximated by the IR rate. Specifically, gene expression quantification is performed twice, first employing the anticipated library strandedness and then utilizing the opposite strandedness, to obtain two gene-by-sample count matrices. Subtracting the counts of the latter from the former generates the corrected gene expression matrix, which can be used for any quantification-based analysis. Though a stranded RNA-seq dataset with gDNA contamination is not available to directly validate our correction method, we hold confidence in its reliability, given the preservation of strandedness of RNA transcripts during stranded library preparation. Such a dataset would serve as invaluable ground truth for evaluating the performance of the three correction methods designed for unstranded RNA-seq datasets, particularly in estimating both false positive and false negative rates in differential gene expression.

In practice, it is crucial to prioritize the removal of gDNA contamination during RNA preparation for RNA-seq. We advocate for the thorough removal of gDNA upfront rather than relying on bioinformatics correction, which should be considered as a last resort. On-column DNase I treatment may not guarantee the complete elimination of DNA due to potential inhibition of the DNase activity by residual lysis solution and other contaminants, leading to incomplete digestion of gDNA. For the efficient removal of gDNA, we recommend employing a combination of the RNAqueous^®^-4PCR Kit and the DNA-free DNase treatment and removal reagents from Ambion (ThermoFisher Scientific, Waltham, MA, USA). This method is effective even for samples with limited quantity, such as 100 cells or 1 mg of tissue [59]. As a gatekeeping measure, RNA samples intended for RNA-seq should undergo examination of gDNA contamination by qPCR without reverse transcription. We suggest utilizing primers targeting ribosomal DNA (rDNA) for this purpose, leveraging their high copy number in genomes for improved sensitivity [65]. If DNase digestion is not feasible, stranded RNA-seq is recommended. PolyA(+) selection-based, unstranded RNA-seq can be considered as the next option, while unstranded RNA-seq with a non-polyA(+)-RNA selection method should be avoided whenever possible. For existing unstranded and stranded RNA-seq data exhibiting apparent gDNA contamination, we recommend employing the “IR%” method and the dedicated approach that utilizes the strandedness information to correct gDNA contamination in differential expression and co-expression analyses, respectively.

## 5. Conclusions

The CleanUpRNAseq package proves highly adept at detecting and rectifying gDNA contamination, thereby significantly enhancing the accuracy of expression quantification and differential expression analysis in RNA-seq data generated through diverse RNA-seq protocols. Given the inherent challenges of gDNA contamination in RNA-seq, this tool should be adopted as standard practice for the post-alignment quality assessment of RNA-seq data. By adopting this approach, the CleanUpRNAseq package has the potential to greatly enhance RNA-seq-based quantification, differential expression analysis, and co-expression analysis. Notably, we have integrated CleanUpRNAseq into OneStopRNAseq [66] for the prompt detection and correction of gDNA contamination in RNA-seq data sourced from GEO or uploaded by users.

## Figures and Tables

**Figure 1 biotech-13-00030-f001:**
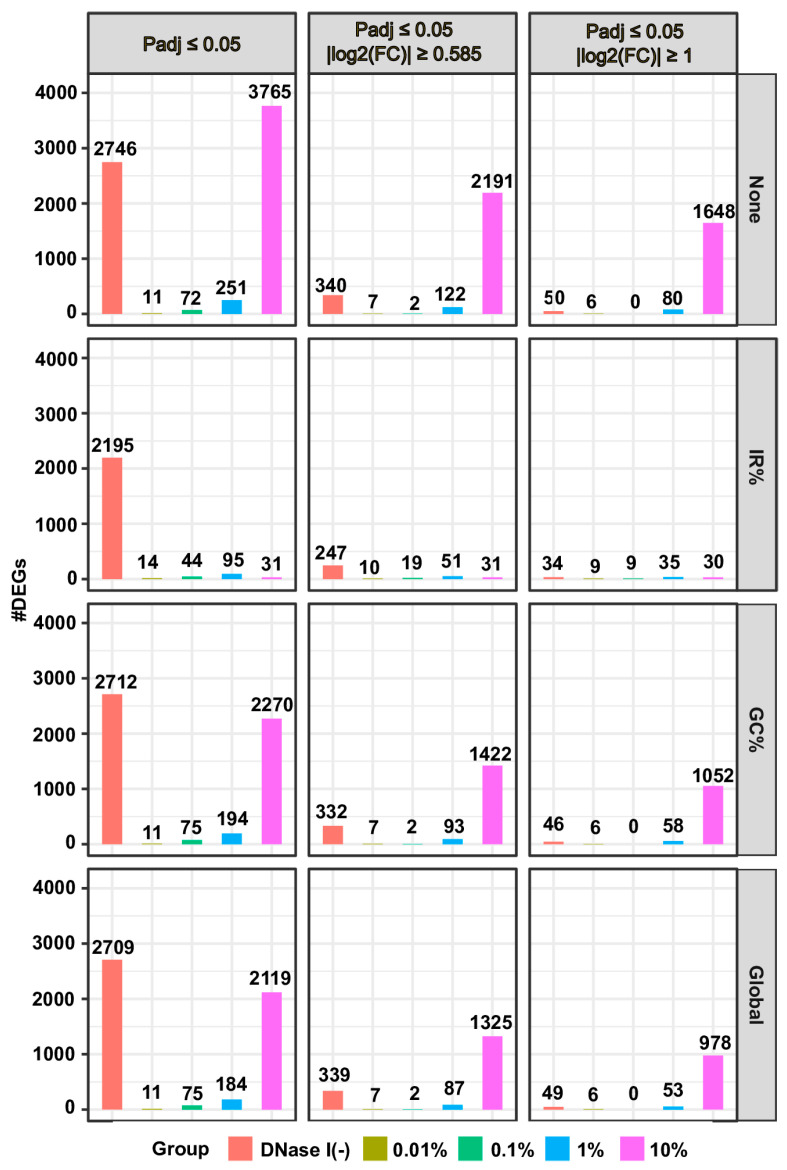
Correction for gDNA contamination reduces false positive DEGs in RNA-seq data generated by the rRNA depletion method. The group of DNase I-treated RNA samples with no added gDNA (Samples B4–B6, Dataset I) served as the control group. Differential expression analyses were performed between this control group and RNA samples untreated with DNase I (DNase I (−)) or DNase I-treated RNA samples with varying levels of added gDNA (0.01%, 0.1%, 1%, or 10%). For the panel labels, “IR%”, “GC%”, and “Global” indicate the respective gDNA contamination correction method applied, while “None” represents the absence of any correction. “None” and “IR%” signify the use of the raw count matrix as the argument of the countData parameter of the DESeqDataSetFromMatrix function, while “GC%” and “Global” denote the utilization of “GC%” and “Global” method-corrected count matrices, respectively. Regarding the design matrix specification, “IR%” utilizes “design = ~group + IR%”, while the other three employ “design = ~group”. To assess the correction effects, three progressively stringent criteria were used for identifying DEGs: adjusted *p*-value ≤ 0.05 (left); adjusted *p*-value ≤ 0.05 and |log_2_(FoldChange)| ≥ 0.585 (middle); adjusted *p*-value ≤ 0.05 and |log_2_(FoldChange)| ≥ 1 (right).

**Figure 2 biotech-13-00030-f002:**
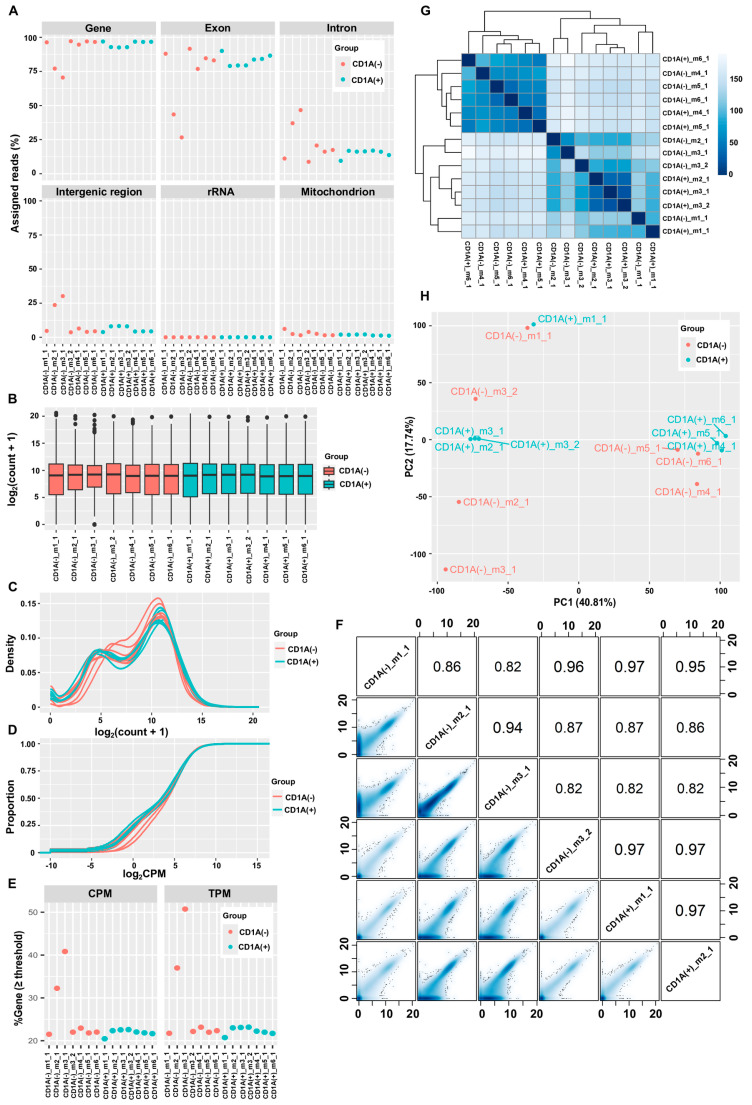
Diagnostic plots reveal gDNA contamination in Dataset II. (**A**) Dot plots showing percentages of reads mapping to different genomic features: genes, exons, introns, intergenic regions, rRNA exons, and the mitochondrial genome. (**B**) Box plots showing sample-level gene expression distributions. The gene-by-sample count matrix was normalized using the DESeq2’s median of ratios method, followed by log-transformation with a pseudocount of 1 added to each value. (**C**) Density plots showing sample-level gene expression distributions. The count matrix was normalized and transformed as in (**B**). (**D**) Empirical cumulative distributions of the sample-level gene expression. The gene-by-sample count matrix was converted to a gene-by-sample CPM matrix using the cpm function of the edgeR package. A pseudocount equal to one-tenth of the minimal CPM value in the matrix was added to each entry in the CPM matrix, followed by log-transformation. The resulting matrix was used to generate the empirical cumulative distributions of sample-level gene expression using the stat_ecdf function of the ggplot2 package. The empirical cumulative distributions of the sample-level gene expression show the proportions of genes (y-axis) with a log_2_CPM equal to or less than any given value (x-axis). (**E**) Dot plots showing percentages of genes with expression levels above one CPM (left) and one TPM (right). (**F**) Smooth scatter plots and Pearson correlation coefficients showing similarities in gene expression profiles between sample pairs. The count matrix was normalized and transformed as in (**B**). Due to space constraints, only a visualization for six samples is presented here, including the two heavily contaminated samples CD1A(−)_m2_1 and CD1A(−)_m3_1. The complete visualization can be found in Appendix A. The *x* and *y* axes show the normalized, log-transformed gene expression. (**G**) Hierarchically clustered heatmap showing Euclidean distances between pairs of gene expression profiles. The gene-by-sample count matrix was normalized as in (**B**) and transformed using the vst function of the DESeq2 package. The heatmap was generated using the pheatmap function of the pheatmap package. The color key shows the scale of distances. (**H**) PCA score plots showing variability among gene expression profiles. The transformed matrix in (**G**) was scaled, centered, and used for PCA with the prcomp function in the base library of R.

**Figure 3 biotech-13-00030-f003:**
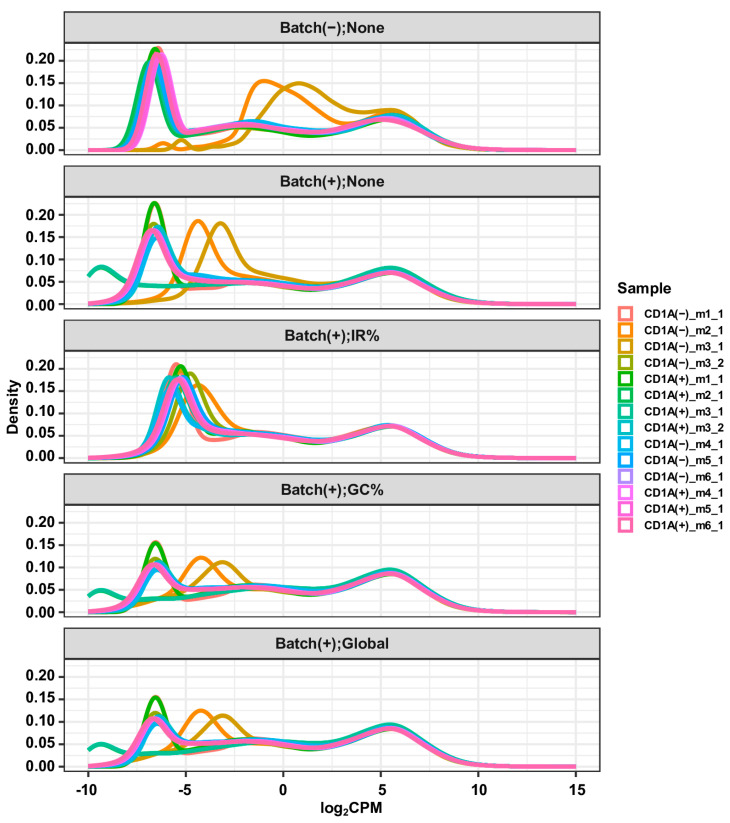
Correction of gDNA contamination and batch effects makes gene expression profiles of CD7^+^CD1A^+^ and CD7^+^CD1A^−^ cells more similar. Density plots show gene expression distributions of RNA-seq Dataset II, which was generated in four batches, as described in the Methods and Materials section and Appendix A. For the panel labels, “Batch(−);None”, “Batch(+);None”, “Batch(+);IR%”, “Batch(+);GC%”, and “Batch(+);Global” indicate whether batch effects and gDNA contamination were corrected, and the respective method applied. “Batch(+)” and “Batch(−)” indicate whether batch effects were corrected or not, respectively. “None” signifies no correction for gDNA contamination, while “IR%”, “GC%”, and “Global” denote the corresponding gDNA contamination correction method applied. “None” and “IR%” denote the use of the raw count matrix, while “GC%” and “Global” indicate the utilization of “GC%” and “Global” method-corrected count matrices, respectively. The corresponding count matrix, whether raw or corrected, was transformed into a log_2_CPM matrix using the voom function of the limma package. These log_2_CPM matrices were used to fit linear models for differential expression analysis using the lmFit function of the limma package, with the model matrices specified as “~group” (Batch(−);None), “~group + batch” (“Batch(+);None”), “~group + batch + IR%” (“Batch(+);IR%”), “~group + batch” (“Batch(+);GC%”), “~group + batch” (“Batch(+);Global”). The estimated parameters for covariates, where appropriate, were used to obtain corrected log_2_CPM matrices, through matrix operations, for visualization.

**Figure 4 biotech-13-00030-f004:**
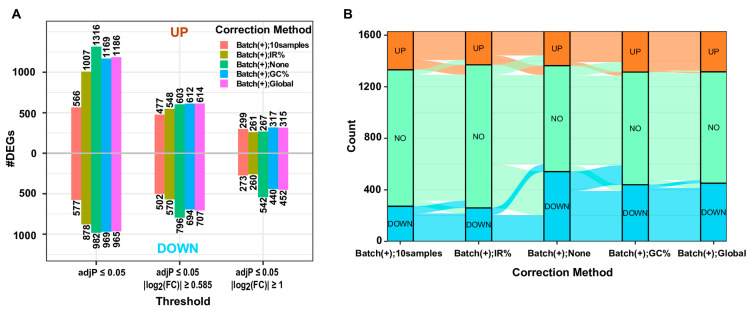
Correction for gDNA contamination reduces DEGs between CD7^+^CD1A^+^ and CD7^+^CD1A^−^ cells. Differential expression analyses were performed between CD7^+^CD1A^+^ and CD7^+^CD1A^−^ cells sorted from PDXs, with the latter serving as the control. Analyses were carried out utilizing all samples, with various gDNA correction methods, if applied, except for the “Batch(+);10samples” analysis, in which the two samples heavily contaminated with gDNA (CD1A(−)_m2_1 and CD1A(−)_m3_1), as well as their paired samples (CD1A(+)_m2_1 and CD1A(+)_m3_1), were excluded. “Batch(+);None”, “Batch(+);10samples”, and “Batch(+);IR%” denote the use of the raw count matrix as the argument of the countData parameter of the DESeqDataSetFromMatrix function, while Batch(+);GC%” and “Batch(+);Global” indicate the utilization of “GC%” and “Global” method-corrected count matrices, respectively. Regarding the design matrix specification, “IR%” utilizes “design = ~ group + batch + IR%”, while the other four employ “design = ~group + batch”. Please refer to the legend of Figure 3 for an explanation of the remaining labels. (**A**) Bar plots showing the numbers of DEGs with various correction methods and three progressively stringent criteria: adjusted *p*-value ≤ 0.05 (**left**); adjusted *p*-value ≤ 0.05 and |log_2_(FoldChange)| ≥ 0.585 (**middle**); adjusted *p*-value ≤ 0.05 and |log_2_(FoldChange)| ≥ 1 (**right**). The numbers of up-regulated DEGs and down-regulated DEGs for specified thresholds are displayed above and below the x-axis, respectively. (**B**) Sankey plots illustrating the status change (UP, NO, or DOWN) of genes before and after gDNA contamination correction using various methods, with a cutoff of adjusted *p*-value ≤ 0.05 and |log_2_(FoldChange)| ≥ 1. UP, NO, and DOWN represent up-regulated DEGs, non-DEGs, and down-regulated DEGs, respectively. To enhance visualization, not all persistent non-DEGs were displayed. Padj refers to the adjusted *p*-value calculated using the “BH” method, and log_2_(FC) represents log_2_(FoldChange).

**Figure 5 biotech-13-00030-f005:**
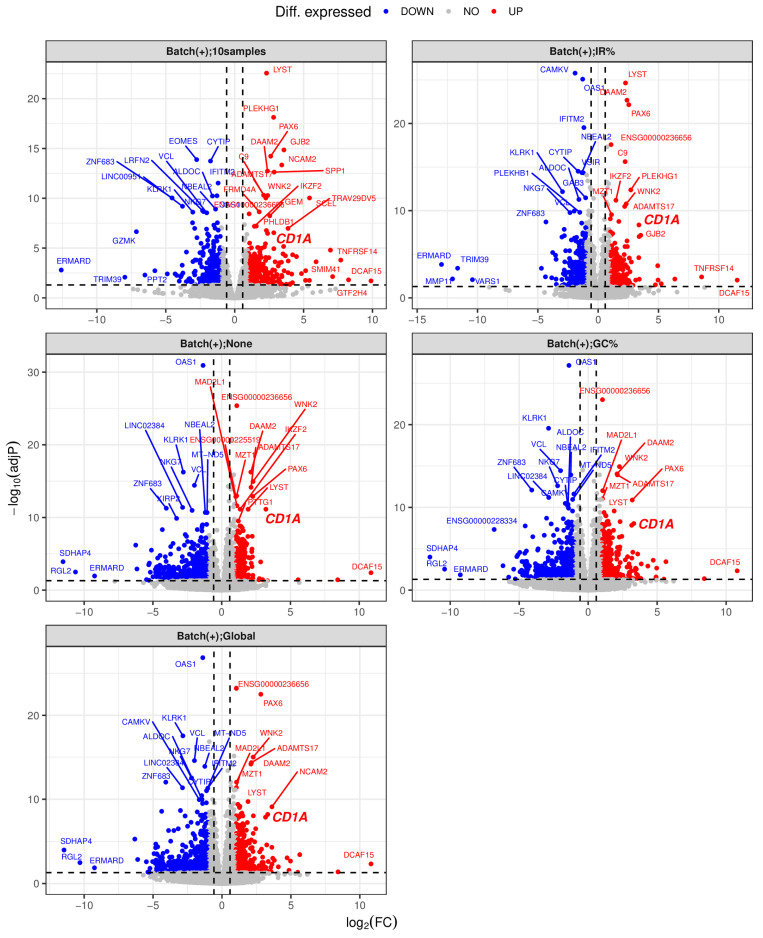
Volcano plots showing DEGs between CD7^+^CD1A^+^ and CD7^+^CD1A^−^ cells before and after gDNA contamination correction. Differential expression analyses and labels were as described in the legends of Figure 3 and Figure 4. Gene were categorized into three groups, including down-regulated DEGs (DOWN, blue), non-DEGs (NO, gray), and up-regulated DEGs (UP, red), with a cutoff of adjusted *p*-value ≤ 0.05 and |log_2_(FoldChange)| ≥ 1. A large portion of the down-regulated genes between CD7^+^CD1A^+^ and CD7^+^CD1A^−^ cells may have been false positives, likely attributable to significant gDNA contamination in two replicates of CD7^+^CD1A^−^ cells, particularly if gDNA contamination correction was inadequate. The “IR%” method demonstrated the most effective reduction in down-regulated DEGs in CD7^+^CD1A^+^ cells compared to CD7^+^CD1A^−^ cells, with the “GC%” and “Global” methods showing similar effectiveness.

## Data Availability

The RNA-seq dataset under accession number HRA001834 (Dataset I) was downloaded from the repository of the Genome Sequence Archive for Human of National Genomics Data Center of China (https://ngdc.cncb.ac.cn/gsa-human/ (accessed on 13 July 2024)). Another RNA-seq dataset (Dataset II, Accession number: GSE260697) was downloaded from the NCBI Gene Expression Omnibus (GEO). Scripts used to analyze the two datasets are available at Git Hub: https://github.com/haibol2016/cleanuprnaseq_data_analysis_scripts/ (accessed on 13 July 2024). The CleanUpRNAseq package is available at Git Hub: https://github.com/haibol2016/CleanUpRNAseq.git (accessed on 13 July 2024). The analyses performed in this study were based on CleanUpRNAseq (RELEASE_0.99.13). It has also been accepted by Bioconductor.

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
