# Peer review of "CleanUpRNAseq: An R/Bioconductor Package for Detecting and Correcting DNA Contamination in RNA-Seq Data"

_biotech, 2024, doi:10.3390/biotech13030030_

Round 1

Reviewer 1 Report

Comments and Suggestions for Authors

This review strongly supports the publication of Liu et al. in BioTech.

  • ClenUpRNAseq: A valuable package for detecting and correcting DNA contamination, particularly in ribosome-minus RNA-seq data.
  • Strengths:
    • Clear and concise presentation of the work.
    • Well-described methods.
    • Clear organization and presentation of figures.
    • High-quality writing.
    • Commendable practice of sharing code on GitHub prior to publication.

          Validation: The authors effectively benchmarked the method using ideal test datasets.

Author Response

This review strongly supports the publication of Liu et al. in BioTech.

  • ClenUpRNAseq: A valuable package for detecting and correcting DNA contamination, particularly in ribosome-minus RNA-seq data.
  • Strengths:
    • Clear and concise presentation of the work.
    • Well-described methods.
    • Clear organization and presentation of figures.
    • High-quality writing.
    • Commendable practice of sharing code on GitHub prior to publication.

          Validation: The authors effectively benchmarked the method using ideal test datasets.

Response: Thank you for your favorable comments regarding our manuscript!

Reviewer 2 Report

Comments and Suggestions for Authors

This paper introduces CleanUpRNAseq, a promising open-source R/Bioconductor package designed to detect and correct for genomic DNA (gDNA) contamination in RNA-seq data. This tool addresses a critical need in the field of RNA sequencing, where gDNA contamination can significantly affect the quality and reliability of results. The authors provide compelling evidence of CleanUpRNAseq's effectiveness in improving post-alignment quality assessment and underscore its potential as a standard practice in RNA-seq workflows.

According to the authors, CleanUpRNAseq is distinguished by its robust ability to identify and mitigate gDNA contamination in RNA-seq datasets. This feature is particularly valuable given the scarcity and high value of RNA samples, making it essential to maximize the utility of the data obtained from these samples. The ability of this tool to clean contaminated data ensures that downstream analyses, such as gene expression quantification and differential expression analysis, are based on accurate and reliable datasets.

The authors advocate the incorporation of CleanUpRNAseq into routine RNA-seq data analysis workflows. By integrating this package, users can significantly improve the accuracy of their analyses. In particular, the inclusion of CleanUpRNAseq in the OneStopRNAseq pipeline is highlighted as a strategic enhancement that ensures comprehensive quality control and improved analytical results.

Overall, this paper presents a compelling and its recommendations for wider adoption and integration into existing pipelines are well founded. Improvements are needed in the presentation of figures, and emphasis should be placed on introducing and discussing non-standard RNAseq applications such as RNA-lipid interactions (see below). Once these issues are addressed, this work will have a lasting impact on the field of RNA-seq data analysis.

Points:

-          The paper should acknowledge the existence of certain RNA-seq methods, such as RNA-lipid interactions described elsewhere (PMID: 37349607), that may not require the same level of contamination correction. Authors should mention this in the introduction or briefly comment on it in the discussion.

-          While the content of the figures is informative, the readability of the graphs, particularly in Figures 1 and 2 (A, B, C, D, F), is compromised by small font sizes. Enlarging the text descriptions and axis labels would greatly improve the clarity and usability of these visual aids.

Author Response

This paper introduces CleanUpRNAseq, a promising open-source R/Bioconductor package designed to detect and correct for genomic DNA (gDNA) contamination in RNA-seq data. This tool addresses a critical need in the field of RNA sequencing, where gDNA contamination can significantly affect the quality and reliability of results. The authors provide compelling evidence of CleanUpRNAseq's effectiveness in improving post-alignment quality assessment and underscore its potential as a standard practice in RNA-seq workflows.

According to the authors, CleanUpRNAseq is distinguished by its robust ability to identify and mitigate gDNA contamination in RNA-seq datasets. This feature is particularly valuable given the scarcity and high value of RNA samples, making it essential to maximize the utility of the data obtained from these samples. The ability of this tool to clean contaminated data ensures that downstream analyses, such as gene expression quantification and differential expression analysis, are based on accurate and reliable datasets.

The authors advocate the incorporation of CleanUpRNAseq into routine RNA-seq data analysis workflows. By integrating this package, users can significantly improve the accuracy of their analyses. In particular, the inclusion of CleanUpRNAseq in the OneStopRNAseq pipeline is highlighted as a strategic enhancement that ensures comprehensive quality control and improved analytical results.

Overall, this paper presents a compelling and its recommendations for wider adoption and integration into existing pipelines are well founded. Improvements are needed in the presentation of figures, and emphasis should be placed on introducing and discussing non-standard RNAseq applications such as RNA-lipid interactions (see below). Once these issues are addressed, this work will have a lasting impact on the field of RNA-seq data analysis.

Points:

-          The paper should acknowledge the existence of certain RNA-seq methods, such as RNA-lipid interactions described elsewhere (PMID: 37349607), that may not require the same level of contamination correction. Authors should mention this in the introduction or briefly comment on it in the discussion.

Response:

Thank you for bringing up the lipid-interacting RNA sequencing method (LIPRNAseq). However, our primary focus remains on addressing potential gDNA contamination in RNA-seq data produced through conventional bulk RNA-seq methods. In cases where gDNA is specifically removed prior to sequencing library preparation in any RNA-seq method, we believe that the impact of gDNA contamination on downstream analyses is minimal.

Regarding the paper (PMID: 37349607), although the authors mentioned that their data is publicly available through NCBI databases (BioSamples and BioProject), we were unable to locate the data during our review. Additionally, the experiments were not replicated. Therefore, without access to the raw data and considering that RNA extracts in LIPRNAseq were treated with DNase I, we cannot ascertain the presence or extent of gDNA contamination in the generated RNA-seq data.

Therefore, we anticipate minimal impact of gDNA contamination on the integrity of data generated by other RNA-seq methods, which typically involve selective removal of gDNA contamination(Lines 676-678). We clarified the terminology of “RNA-seq” in the Introduction (Lines 36-37) by emphasizing that “(hereafter referred to as conventional bulk RNA-seq unless stated otherwise)”, and acknowledged the presence of numerous other RNA-seq methods in the Discussion (Lines 658-662), referencing the paper (PMID: 37349607).

-          While the content of the figures is informative, the readability of the graphs, particularly in Figures 1 and 2 (A, B, C, D, F), is compromised by small font sizes. Enlarging the text descriptions and axis labels would greatly improve the clarity and usability of these visual aids.

Response:

We increased the font sizes of Figures 1-4 to improve label readability and included high-resolution PDF versions of all figures as supplementary materials.

We sincerely appreciate the time and effort the reviewers dedicated to carefully reading our manuscript. We are grateful for their positive feedback and valuable suggestions!

We have addressed all of the comments below, and our responses are indicated in bold italics.

Reviewer 3 Report

Comments and Suggestions for Authors

The manuscript titled “CleanUpRNAseq: an R/Bioconductor package for detecting and correcting for DNA contamination in RNA-seq data” by Zhu, developed a tool called CleanUpRNAseq to detect and correct gDNA contamination, offering three methods for unstranded and one for stranded RNA-seq data. It is a valuable tool for RNA-seq quality assessment and should be integrated into routie workflows to improve gene expression analysis accuracy. This is a novel aspect in a field of intense research. Still some issues need to be clarified, as listed below, before the manuscript can be accepted for publication in BioTech.

1.      L110, the authors mention two existing tools for addressing gDNA contamination, namely gDNAx and SeqMonk. Compared to these tools, how does CleanUpRNAseq achieve improvements in accuracy? Please provide detailed explanations and evidence to support the claims of enhanced accuracy with CleanUpRNAseq, particularly in terms of its correction methods and validation results.

2.      L158, the author mention the GC% method for accounting for GC content bias effect on read distribution across genomic regions. Please explain the statistical model and assumptions behind the GC% method, and how does GC% method handle variations in GC content across different regions and the potential biases introduced? Providing a detailed explanation of the methodology and its effectiveness in mitigating GC content bias would be helpful.

3.      Can the package differentiate between gDNA contamination and other sources of error noise in RNA-seq data? Please provide details on the methodologies used to distinguish gDNA contamination from other types of errors or noise, and discuss the effectiveness and limitations of these approaches.

4.      Do the authors have any plans for the future development and application of CleanUpRNAseq? What aspects need to be improved during its implementation? Please provide details on how the tool may evolve and any areas identified for enhancement to increase its effectiveness and user-friendliness.

Author Response

The manuscript titled “CleanUpRNAseq: an R/Bioconductor package for detecting and correcting for DNA contamination in RNA-seq data” by Zhu, developed a tool called CleanUpRNAseq to detect and correct gDNA contamination, offering three methods for unstranded and one for stranded RNA-seq data. It is a valuable tool for RNA-seq quality assessment and should be integrated into routie workflows to improve gene expression analysis accuracy. This is a novel aspect in a field of intense research. Still some issues need to be clarified, as listed below, before the manuscript can be accepted for publication in BioTech.

  1. L110, the authors mention two existing tools for addressing gDNA contamination, namely gDNAx and SeqMonk. Compared to these tools, how does CleanUpRNAseq achieve improvements in accuracy? Please provide detailed explanations and evidence to support the claims of enhanced accuracy with CleanUpRNAseq, particularly in terms of its correction methods and validation results.

Response:

Thank you for your great question! As mentioned in lines 113-115, while gDNAx can be seamlessly integrated into automated RNA-seq pipelines, its current correction method is rudimentary, primarily filtering intronic and intergenic reads. However, it inadequately addresses gDNA reads mapped to exons, thus lacking functionality for correcting gDNA contamination effectively. For details, please refer to the gDNAx Vignette (https://www.bioconductor.org/packages/release/bioc/vignettes/gDNAx/inst/doc/gDNAx.html), specifically Section 3 on removing gDNA contamination, and view the source code at https://github.com/rcastelo/gDNAx/blob/devel/R/filterBAMtx.R#L164-L191. gDNAx employs the filterBam function from the RSamtools package to filter aligned reads based on user-defined SAM flags. However, this method does not remove reads aligned to exonic regions, as these can originate from both genomic DNA contamination and RNA. Therefore, relying solely on a GTF file and BAM file, gDNAx cannot effectively remove reads originating from gDNA contamination but aligned to exonic regions. Hence, we did not compare it with our methods.

 At lines 156-159, we highlight that “More importantly, CleanUpRNAseq offers three correction methods for unstranded RNA-seq data:  the “Global” method assuming a uniform distribution of gDNA contamination, which is a re-implementation of the SeqMonk method in R, …”. Therefore, in our comparison of the three correction methods for unstranded RNA-seq datasets, we have already evaluated the SeqMonk method alongside our other approaches.

  1. L158, the author mention the GC% method for accounting for GC content bias effect on read distribution across genomic regions. Please explain the statistical model and assumptions behind the GC% method, and how does GC% method handle variations in GC content across different regions and the potential biases introduced? Providing a detailed explanation of the methodology and its effectiveness in mitigating GC content bias would be helpful.

Response:

We have detailed the operation of the GC% method as follows (see Lines 223-237): “For the “GC%” method, GC content of individual intergenic regions and genes are calculated using the calculate_region_gc and calculate_gene_gc functions, respectively. For each sample, a loess regression model is fitted with the FPB of intergenic regions with non-zero counts as the response variable and GC content as the independent variable. GC content is divided into 20 equal-width bins, and intergenic regions are assigned to their corresponding GC bins based on their GC content. Coverages in FPB of intergenic regions within each GC bin are predicted by the fitted loess regression model. The median coverage in FPB of intergenic regions within each GC bin serves as GC bin-specific estimate of gDNA contamination. For a given sample, GC content of individual genes is similarly binned as for intergenic region. Per-gene contamination is calculated by multiplying the GC bin-specific estimate of gDNA contamination for that sample and the length of the gene output by the salmon_res function. For both methods, per-gene gDNA contamination is subtracted from the raw per-gene counts, resulting in a gDNA contamination-corrected count matrix. These methods are implemented as the global_correction and gc_bias_correction functions.”

Previous studies, such as those described in https://academic.oup.com/nar/article/40/10/e72/2411059, have demonstrated the impact of GC content bias on fragment counts in high-throughput DNA-seq. Our 'GC%' method assumes a uniform distribution of reads originating from gDNA contamination across narrow GC-content bins. The relationship between GC content and fragment count is more pronounced at the fragment level. We simplify by calculating GC content for each intergenic region and metagene, potentially affecting the performance of the 'GC%' method.

The efficacy of the 'GC%' method compared to the other two methods was discussed at Lines 437-440, “For RNA-seq data generated using the rRNA-depletion method, applying the “IR%” correction method led to a significant decrease in the numbers of DEGs between the control group and samples with 1% or 10% of gDNA added, while the “GC%” and “Global” correction methods showed moderate but comparable effects (Figures 1 and S7)”, and similarly at Lines 533-536, “After examining density plots of gene expression profiles before and after correction with the three methods (Figure 3), we concluded that the “IR%” method performed best, followed by the “GC%” and “Global” methods, which exhibited similar performance.”

  1. Can the package differentiate between gDNA contamination and other sources of error/noise in RNA-seq data? Please provide details on the methodologies used to distinguish gDNA contamination from other types of errors or noise, and discuss the effectiveness and limitations of these approaches.

Response:

CleanUpRNAseq primarily utilizes reads mapped to intergenic regions across a genome to detect and correct gDNA contamination in unstranded RNA-seq data. Any errors or noise that affect the estimation of counts or percentages of reads mapped to intergenic regions can inadvertently impact the accuracy of gDNA contamination estimation and subsequent correction. Currently, we employ a static GTF file to define intergenic regions for a reference genome. However, the transcriptome is dynamic and complex, with not all genes in the GTF file being expressed in a given sample, and not all expressed transcripts being accurately annotated. Therefore, our current implementation may either overestimate or underestimate the level of gDNA contamination, depending on the GTF file used. Ideally, a sample-specific set of intergenic regions would provide a more precise estimate of gDNA contamination.

Additionally, CleanUpRNAseq does not account for errors or noise introduced by RNA-seq assays (such as sampling bias, PCR amplification bias, and sequencing bias) and bioinformatics analysis (including mapping bias/error and read summarization). All three correction methods for unstranded RNA-seq data rely on statistical models and assumptions: the "Global" method assumes a uniform distribution of reads from gDNA contamination, the "GC" method assumes a uniform distribution within specified GC-content bins, and the "IR%" method assumes a linear relationship between log(normalized read count) and IR%.

In our discussion at Lines 703-735, we evaluate the effectiveness and limitations of each method. We acknowledge that current statistical methods cannot completely and accurately correct for gDNA contamination in RNA-seq data, although they can mitigate its effects. At Lines 736-755, we provide practical advice for minimizing gDNA contamination experimentally in RNA-seq experiments and recommend strategies for data analysis when gDNA contamination is present in RNA-seq data.

  1. Do the authors have any plans for the future development and application of CleanUpRNAseq? What aspects need to be improved during its implementation? Please provide details on how the tool may evolve and any areas identified for enhancement to increase its effectiveness and user-friendliness.

Response:

We integrated the CleanUpRNAseq package into our established OneStopRNAseq pipeline (see Lines 764-766), a user-friendly web application designed for comprehensive and efficient RNA-Seq data analysis by both bioinformaticians and biologists. Details can be found at https://pubmed.ncbi.nlm.nih.gov/33023248/ and https://mccb.umassmed.edu/OneStopRNAseq/. While there may be room for improvement in the GC% method, we do not anticipate it will surpass the IR% method in performance. Concurrently, the CleanUpRNAseq package has been submitted for review to Bioconductor (https://www.bioconductor.org/) and is currently under evaluation (https://github.com/Bioconductor/Contributions/issues/3442#issuecomment-2182816902). Upon successful review, it will be formally accepted by Bioconductor and updated biannually, aligning with the evolution of other Bioconductor packages and user-requested features.

We sincerely appreciate the time and effort the reviewers dedicated to carefully reading our manuscript. We are grateful for their positive feedback and valuable suggestions!